# Knowledge, Attitudes, and Practices of Sex Workers of Three South African Towns towards Female Condom Use and Contraceptives

**DOI:** 10.3390/healthcare11091271

**Published:** 2023-04-28

**Authors:** Noluvuyo Sitonga, Sibusiso C. Nomatshila, Mahlane Phalane, Wezile W. Chitha, Sikhumbuzo A. Mabunda

**Affiliations:** 1Department of Public Health, Walter Sisulu University, Mthatha 5117, South Africa; 2Mpumalanga Department of Health, Witbank Hospital, Emalahleni 1035, South Africa; 3Health Systems Enablement and Innovation Unit, University of the Witwatersrand, Johannesburg 2193, South Africa; 4School of Population Health, University of New South Wales, Sydney 2052, Australia; 5The George Institute for Global Health and Research, University of New South Wales, Sydney 2042, Australia

**Keywords:** female and (sex workers or sex-workers), female condom, contraceptives, women’s health

## Abstract

Female sex workers are a marginalized and highly vulnerable population who are at risk of HIV and other sexually transmitted diseases, harassment, and unplanned pregnancies. Various female condoms are available to mitigate the severity of the consequences of their work. However, little is known about the acceptability and usage of female condoms and contraceptives among sex workers in small South African towns. This descriptive cross-sectional study of conveniently selected sex workers explored the acceptability and usage of female condoms and contraceptives among sex workers in South Africa using validated questionnaires. The data were analyzed using STATA 14.1. The 95% confidence interval is used for precision, and a *p*-value ≤ 0.05 is considered significant. Out of 69 female-only participants, 49.3% were unemployed, 53.6% were cohabiting, and 30.4% were HIV positive. The median age of entry into sex work was 16 years old. Participants reported use of condoms in their last 3 sexual encounters (62.3%), preference of Implanon for contraception (52.2%), barriers to condom use (81.2%), condoms not being accepted by clients (63.8%), being difficult to insert (37.7%), and being unattractive (18.8%). Participants who reported barriers to condom use were 90% more likely to have adequate knowledge than those who did not (PR = 1.9; *p*-value < 0.0001). Knowledge of condom use was an important factor in determining knowledge of barriers to their use. Reasons for sex work, sex workers’ perceptions, and clients’ preferences negatively affect the rate of condom use. Sex worker empowerment, community education, and effective marketing of female condoms require strengthening.

## 1. Background

Sexually transmitted infections (STIs), including HIV/AIDS, are a major public health burden in South Africa [1,2]. The epidemic is complex and thought to be influenced by a number of factors, including biological, behavioural, societal, and structural factors [1]. Although the contraceptive utilisation rate is high in South Africa, at 64% among sexually active women, unplanned and teenage pregnancies are an on-going problem [3,4]. Many South Africans are believed to be using condoms for HIV prophylaxis, but there are challenges with the use of condoms in certain communities [1,5,6,7,8,9,10].

The South African government supplies free male and female condoms to the population and has positions in numerous areas considered hotspots. This includes hotels, shops, taverns, health facilities, brothels, and every place with a dense population at the same time. One of the biggest problems is that even if condoms are used, they are not used consistently, especially in long-term relationships and among those who engage in high-risk sexual practices, such as sex workers [11]. Much of the high HIV prevalence in South Africa is attributable to the inability of women to negotiate for safer sexual practices, often because of age disparities or financial dependence [11,12]. Lack of female-controlled prevention methods also plays a significant role in a woman’s HIV risk [11,12].

Male condom compliance requires cooperation from the woman’s male partner(s), something that is not always possible in abusive relationships [8,11,12]. Another option to prevent STIs, HIV, and unplanned pregnancies, that is, in women’s control, is the use of female condoms [5,8]. This is both liberating and empowering for the woman, as she is in control of the situation and can practice safer sex if she wants to [8,11,12]. The literature suggest that women’s empowerment and strategies such as the active promotion of female condom use can play a huge role in addressing challenges such as the high rate of STIs, teenage pregnancy, and HIV/AIDS [8,11,12,13].

Female condom use in Africa is realistic, and it provides women with more independent protection [13]. It is an alternative that is in the woman’s control, with less need to rely on the male partner’s cooperation or negotiation skills [14]. However, despite the known effectiveness of female condoms in preventing STIs (and thus reducing their prevalence) among sex workers and women in general, there is low uptake among women, including female health workers [8,14].

The acceptability of the female condom among women faces two obstacles: the reaction of the woman’s regular partner and attitudes towards the device itself (appearance, difficulties, or uneasiness concerning its use) [12,13]. It has, however, also been established that the use of a female condom may cause more stigma and challenges for women [15].

As earlier highlighted, sex workers are often at a higher risk of HIV [8,11,15]. They could benefit from the increased promotion and accessibility of female condoms, as it has been shown that an increase in female condom promotion is positively correlated with an increase in female condom uptake among sex workers in Thailand and Madagascar [8]. Equipped with the correct knowledge, sex workers could then also be recruited as peer educators and advocates of safe sexual practices during their trade [16].

In Calcutta, India, the Sonagachi Project employed sex workers as peer educators to distribute condoms, advise peer sex workers on where they can get health services, and disseminate information promoting behavioural change [17]. The 59% rise in condom use in the same period can be attributed to this collaborative model [16,17]. Other countries’ sex worker advocacy organisations, such as South Africa’s Sex Workers Education and Advocacy Task Force (SWEAT), have adapted the same model of peer education to their context [18].

With no studies reporting on sex work and female condom use in most parts of South Africa, little is known about the acceptability and usage of the female condom among sex workers. This is despite the fact that South African female sex workers are known to have a high HIV prevalence and incidence and are responsible for a significant role in the transmission of HIV [19]. This is because they are known to have unprotected sex with their romantic partners and some of their clients [19,20]. Furthermore, because sex work is illegal in South Africa, advocacy for their sexual and reproductive health is limited [21,22].

This study therefore aimed to determine the knowledge, attitudes, and practices of Grahamstown, Rustenburg, and Brits female sex workers on the use of female condoms and contraceptives. This descriptive study also explores factors that could have informed their knowledge, attitudes, and practices. This study will generate new ideas and new strategies that can be implemented to promote female condom use among women, suited to their needs.

## 2. Methods

### 2.1. Study Design

The study is a quantitative design that surveyed participants over 10 days in 2018. Where it is not explicitly stated to be a female condom, the phrase “condom” refers to condom use in general.

### 2.2. Study Setting

The North-West (NW) and Eastern Cape (EC) provinces are two of South Africa’s nine (9) provinces located in the north-west and south-east of the country, respectively. The study was conducted in two brothels in Brits and Rustenburg (NW) and in two brothels in Joza township in Grahamstown (EC) between 20–29 September 2018. These are small towns in predominantly rural provinces with similarities in that they have high proportions of truck rest stations and migrant labourers [23]. As mining towns, both Brits and Rustenburg have high proportions of migrant male workers who work far from their wives and thus serve as a good market for female sex workers.

### 2.3. Population and Sampling

The target population for this research involved female sex workers of all ages trading at Brits and Rustenburg, under the jurisdiction of Bojanala District in the NW province, and Grahamstown, under the jurisdiction of Cacadu District in the EC province.

The principal investigator (PI) met with all the respondents at their risk reduction workshops (RRW), held on selected days of the week. This setting made it easier since they were not focusing on clients at the time. Seventy participants were offered consent forms on three occasions during the workshop, and all were returned signed as all participants were willing to participate. Surveys were undertaken in a private room within the center where none of the other participants could overhear.

### 2.4. Measurements

A researcher-administered questionnaire that was translated into isiXhosa and Setswana was used to collect demographic information, knowledge, beliefs, attitudes, and practices regarding condom use (mostly the female condom), and questions on sexual activity. The latter questions (on sexual activity) were adapted from the youth risk behaviour survey [24] and also incorporated common themes (sexual risk reduction, condom promotion, access, cost, and availability) found in literature [8,11,12,14]. Nine questions were used to assess knowledge of the female condom and contraceptives, the availability, costs, and effectiveness of the former in preventing HIV and STIs in general, and contraceptive options available in the South African public health sector. The content validity was reviewed by two experts (a health promoter and a public health medicine specialist), and there was 100% agreement on clarity, and the content validity index was 1.0. A knowledge score of at least 50% was considered adequate. The views of participants on female condoms were assessed for the best possible view on four options that were not necessarily mutually exclusive to ascertain the commonly expressed view. Perceptions of female condom use were assessed using a 3-item Likert scale (disagree, neutral, agree), where neutral was equivalent to being unsure and/or never having used a female condom before. An exception is the perception of access to female condoms, which also uses a 3-item Likert scale (very difficult, somewhat difficult, and not difficult). The translation of the questionnaire and the presence of a single interviewer for all participants enhanced the reliability of the study’s findings.

### 2.5. Data Management and Statistical Analysis

All variables were captured and coded in Microsoft Excel 2013 and exported to Stata 14.1 for analysis. The numerical data were explored using the Shapiro–Wilk test. While numerical data that were normally distributed (age of participants and the age of entry into sex work) are summarised using the mean, standard deviation (SD), and range, numerical data that were not normally distributed (age of sex debut and the average number of daily sexual clients) are reported using the median and interquartile range (IQR). The two-sample t-test for equal variances was used to test the equality of two means by province, where numerical data were normally distributed, and the Wilcoxon sum rank test (Mann–Whitney U test) was used to test for the equality of two medians if data were not normally distributed.

Categorical variables are presented using frequency tables, percentages, and graphs. Two proportions are compared using the two-sample t-test of proportion. Two numerical variables are compared using the Spearman’s correlation. Simple linear regression is used to compare two different associations of knowledge (age in years and the average number of daily sexual clients). Binomial logistic regression is used for bivariate associations of knowledge for the overall population. The prevalence ratio (PR) is a measure of the association of knowledge. The 95% confidence interval (95% CI) is used to estimate the precision of estimates. The level of significance was a *p*-value ≤ 0.05.

### 2.6. Ethics and Legal Considerations

The Walter Sisulu University Human Ethics and Biosafety Committee granted ethical clearance and approval for the study to be conducted with an ethics approval number (HREC: 005/2019). Each participant gave informed consent; confidentiality was maintained, abiding by the four ethical principles of autonomy, beneficence, non-maleficence, and justice. Participation was completely voluntary without a promise of financial and/or personal incentives.

## 3. Results

A total of 70 participants were interviewed, but one participant from the North-West province was excluded due to inconsistent information on pregnancies and three other variables. As a result, only 69 participants are included in the final analysis, of which 21 (30.4%) and 48 (69.6%) were from the Eastern Cape (EC) and North-West (NW) provinces, respectively. The demographic characteristics are shown in Table 1. On average, participants were 32 years old (sd = 7.2, range = 18–46); the youngest age of entry into sex work was 16 years, and the average age of entry was 22.8 years (mean = 22.8; range = 16.0–35.0).

More than half of the participants (53.6%) were cohabiting; 15.9% had multiple sexual partners; 82.6% had at least a matric as the highest level of education; and 21.7% had a tertiary qualification. Thirty-five (50.7%) had a job ranging from being a peer educator (26.1%), being a cashier or an intern (7.3%), administrative or general assistant work (4.3%), and volunteering as a police reservist (1.4%).

All participants knew their HIV status, and there was a prevalence of 30.4% (95% CI: 20.5–42.5), which comprised 19.0% and 35.4% of participants from the Eastern Cape and North-West provinces, respectively. Implanon was the most commonly used contraceptive (52.2%), followed by 34.8% who were on injectable contraceptives, and this trend was a reflection of the picture in the two provinces.

The first sexual encounter was voluntary for almost two-thirds of the participants (65.2%, 45/69); 92.8% of participants reported to have begun sex work due to poverty or unemployment; and 89.9% of participants had been pregnant before (Table 2). Participants from both provinces reported that they had a minimum of four daily sexual clients (median = 6.0; IQR = 5.0–8.0).

Overall, 68.1% of participants had either been pregnant once or twice, and 22.0% had been pregnant 3 to 5 times. Only 20.3% of participants reported ever having an abortion. Twenty-eight participants (40.6%) had a single child, 30.4% had two, and 15.9% had three or more children (Table 2).

Condom use in the most recent sexual encounter was reported by 82.6% of participants (Table 3). Even though condoms were reported to be advantageous by most participants, 81.2% of participants reported barriers to condom use. Such barriers included nonacceptance by clients, resulting in a negative impact on their income (63.8%). In other instances, clients refused to use a condom, and this was reported by 36.2%.

Whereas 30.4% of participants found female condoms to be a useful preventative method, 49.3% found female condoms to be uncomfortable (Table 4). Whereas 89.9% of participants had not used a condom in the past 3-months, 17.4% reported having used a female condom at least once during the course of the most recent twelve months.

The data in Table 5 show further perceptions of the female condom. Only 8.7% felt female condoms were easy to insert, only 7.3% felt they enhanced pleasure, and 81.2% confirmed that they were adequately promoted.

A total of 65.2% of participants reported having consumed alcohol during their most recent sexual encounter. Of all the participants, only one reported drug use before the most recent intercourse.

Adequate knowledge was attained by 76.2% EC participants and 29.2% NW participants, respectively. Overall, those who reported barriers to condom use were 6.7 times more likely to have adequate knowledge than those who did not, and this was statistically significant (PR = 6.7; 95%CI: 1.01–45.0; *p*-value = 0.004). Furthermore (Table 6), EC participants were 2.6 times more likely to have adequate knowledge than NW participants, and this was statistically significant (PR = 2.6; 95%CI: 1.6–4.3; *p*-value = 0.003). There was no statistically significant association between HIV status and the level of knowledge (PR = 0.6; 95%CI: 0.3–1.2; *p*-value = 0.099).

A 1-year increase in age led to a 0.4% reduction in knowledge score, which was statistically significant (*p*-value = 0.032); despite this, however, only 6.6% of the variation in knowledge score could be attributed to the linear relationship it had with age (R^2^ = 6.6%) (Table 7 and Figure 1). Similarly, the addition of a single sexual client resulted in a 1.9% reduction in knowledge score, and this was also statistically significant (*p*-value = 0.002); as with age, only 13.3% of the variability in knowledge score could be attributed to its linear relationship with the average number of daily sexual clients (R^2^ = 13.3%). None of these associations were statistically significant when stratified by province.

Figure 1 further illustrates the knowledge score for the sex workers in South Africa.

## 4. Discussion

This study sought to understand the knowledge, attitudes, and practices of sex workers towards female condoms and contraceptive use in the South African context.

One of the most critical but under-valued strategies for reducing HIV incidence, other sexually transmitted diseases, and unwanted pregnancies is understanding high-risk populations and their reasons for uptake or failure to utilise interventions to help them. The understanding of the baseline knowledge should trigger behaviour change, habits (occupational practices, alcohol use, multiple sexual partners, etc.), perceived threats, perceived susceptibility to an adverse outcome (e.g., HIV infection, loss of income, etc.), and perceived benefits of behaviour change [25].

In the absence of government-driven programs for sex workers in South Africa, this study therefore adds value to the paucity of literature in this area, not only to inform the design of interventions but also to help find alternative methods of engaging stakeholders that could extend beyond female sex workers in the design of health interventions [16].

Participants in this study reported having begun sex work due to poverty or unemployment, even those with tertiary qualifications. This is consistent with previous UNAIDS findings, which reported that some individuals choose sex work as an occupation, but for some communities, it remains a means of survival, with as many as 86% of Canadian female and child sex workers from indigenous communities having a history of poverty and homelessness [26]. Other factors reported to sex work include a lack of education and/or employment opportunities, marginalisation, addictions, and mental illness [26]. This often affects the younger population, which is still in its prime. A peri-suburban South African study reported a median age of 31 years among sex workers in Soweto [27].

It is impressive to note that more than 80% of the participants who were only females had at least a matric level of education, which puts them above average when compared to any general 25-year-old South African women living in urban and peri-urban areas, whose high-school education attainment was measured at 68.2% [28]. In South Africa there is a high rate of unemployment, with graduates struggling to find jobs. Poverty and unemployment are the most contributing factors. That is the reason why we see young people with matric being in the sex work industry: it is because of poverty and hunger. This also contradicts the association of sex work with a lack of education as seen in other societies elsewhere in the world [26]. Individuals with such a level of education are therefore expected to grasp health promotion with ease if their knowledge is to be enhanced [8,23].

Even though the HIV prevalence of 30.4% is higher than the South African adult population prevalence of 20.4% reported for 2018 [2], it is slightly lower than the South African antenatal HIV prevalence of 30.8% reported in 2015 [2,29]. The HIV prevalence is also far lower than the estimated HIV sex-worker prevalence reported by the United Nations for 2018 of 57.7% [2]. In a study by Coetzee et al., an HIV prevalence of 39.7% was reported in a study population of sex workers in Cape Town, 53.5% in Durban, and 71.8% in Johannesburg [21,27]. With such a high prevalence, it is therefore highly critical for sex workers to protect themselves, their clients, and/or intimate partners against STI (including HIV) infection or re-infection by using dual methods of protection (i.e., condoms and other proven preventive measures). However, condoms were not found to be used consistently in this study, either as a result of non-acceptability by clients, perceptions of the sex workers who lose income, or ineffective marketing approaches. Qualitative data supported survey findings [28] on the inconsistency of condom use resonate with findings from this study where participants opted against condom use with clients for higher payment, substance use clouding their judgment, and the inability to negotiate safer sexual practices with spouses for issues related to trust and fear of sexual violence or force from clients or partners [28]. The disadvantages of condom use raised in this study are a common finding in the literature [20]. Opportunity costs for condom use included the fact that not using condoms had a negative impact on their income, as some clients would either leave or offer to pay less if a condom was used [20]. In other settings, sex workers have reported preference for the female condom as they could have it on before meeting a client, thus eliminating the need to negotiate condom use [8,14,20].

A further complication of unprotected sex and the consequent unplanned pregnancies is the risk associated with abortion (often in the informal sector due to being stigmatized) [20]. Even though the number of women with a previous abortion accounted for a fifth (20.3%) of all the participants and is lower than other previous African reports of between 22 and 86%, it is still of concern [20]. This suggests a lack of use of other family planning services available.

Implanon and injectable contraceptives were the most commonly used contraceptives by 52.2% and 34.8% of participants, respectively, suggesting preferences for medium-term (mostly 3-months) to long-term (5-years) contraceptives. It is consistent with other literature findings where sex workers have a lower tendency for oral contraceptives as they require daily intervention, which could result in poor compliance [8,20]. In contrast, though, some female sex workers reject injectable contraceptives as they are considered bad for business due to their associated dizziness, nausea, and menorrhagia, or extended vaginal bleeding [20].

In previous Kenyan studies [20,28], participants preferred effective medium- or long-term contraceptives such as injectable contraception or an implant [20,28]. The individual circumstances of a sex worker often interfered with compliance and the correct use of other methods [20,28]. Risky behaviours, such as being drunk, were some of the common reasons associated with poor condom and contraceptive compliance [20]. Injectable contraceptives, Implanon, and intrauterine contraceptive devices are also beneficial as contraceptives for female sex workers who are raped or those who cannot negotiate safe sex [20,28]. Furthermore, condoms could also tear, thus the need for the additional contraceptive measure [20,28].

Even though adequate knowledge was attained by only 43.5% of the participants, those with adequate knowledge were 90% more likely to report opportunity costs associated with condom use. By inference, sex workers will engage in unprotected sex not due to a lack of knowledge but often due to the cost of not giving the client what he prefers. This also suggests that health promotion strategies and messaging are not reaching their target audience, suggesting the need for a change of tactics for better impact.

Even though minimised, this study is not without limitations. It is not, however, anticipated that these could have given rise to different outcomes. Firstly, there is a selection bias in that the participants were recruited from a risk reduction workshop (a controlled environment) and are therefore likely to be more knowledgeable about safer practices than the general population of sex workers.

Secondly, the use of a researcher-administered questionnaire could have led to a social desirability bias. This was limited by the asking of follow-up questions and the rephrasing of some questions in a different section of the questionnaire to assess reliability. As a result, a participant with inconsistent responses was eliminated. Even though the findings are not generalisable for all South African sex workers due to the small sample size, the study has certainly identified the health needs of this marginalised population, as the findings are internally valid and could be the basis for more detailed deductive qualitative studies and prospective quantitative studies. Furthermore, the study has highlighted the importance of increasing contraceptive uptake and the need to promote female condom acceptability and availability among female sex workers.

## 5. Limitations

Due to the nature of the work of the participants, it is often difficult to find them in situ. Subsequently, the obtained population size may not be representative. This study also could not investigate in-depth the events leading to the start of sex work. Paucity in literature has also limited the authors’ ability to obtain adequate and recent literature.

## 6. Conclusions

This study has confirmed the low acceptability of female condoms, as manifested by the low usage of female condoms. Though adequately marketed, the effectiveness of marketing on effectiveness and efficacy can be linked to low use, negative perception by sex workers, and unacceptability by clients. Additionally, poverty and high unemployment rates remain challenges in facilitating decisions to engage in sex work. Knowledge of condom use led to a better understanding of barriers to condom use. It becomes imperative to strengthen the approaches used to market condoms to members of the community and improve access in order to improve attitudes and efficacy in their use. Furthermore, the government needs an inclusive approach to dealing with sex work and its associated risks.

## Figures and Tables

**Figure 1 healthcare-11-01271-f001:**
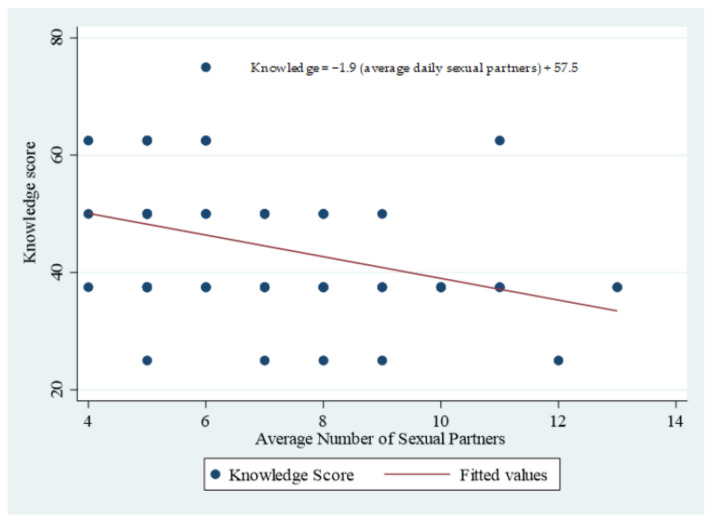
Knowledge Score with the average number of daily sexual partners/clients by South African sex workers.

**Table 1 healthcare-11-01271-t001:** Demographic characteristics of participants living in South Africa.

Province	n (%)	
Eastern Cape	21	(30.4)
North-West	48	(69.6)
Age, years; mean ± sd (Range)	32.0 ± 7.2	(18.0–46.0)
Age of entry into sex work, years; mean ± sd (Range)	22.8 ± 4.4	(16.0–35.0)
Age of sex debut, years; median (p25–p75)	15.0	(15.0–17.0)
Average number of daily sexual clients; median (p25–p75)	6.0	(5.0–8.0)
Marital status; n (%)		
Single	25	(36.2)
Married	3	(4.4)
Divorced	3	(4.4)
Separated	1	(1.5)
Cohabiting	37	(53.6)
Relationship status; n (%)		
One heterosexual partner	51	(73.9)
Multiple heterosexual partners	11	(15.9)
No heterosexual relationships	7	(10.1)
Highest level of education completed; n (%)		
Primary School	3	(4.4)
Grade 10	3	(4.4)
Grade 11	6	(8.7)
Grade 12	42	(60.9)
Tertiary qualification	15	(21.7)
Current job; n (%)		
Administrative assistant	3	(4.4)
Cashier	5	(7.3)
General assistant	3	(4.4)
Police reservist	1	(1.5)
Intern	5	(7.3)
Peer educator	18	(26.1)
Unemployed	34	(49.3)
HIV status; n (%)		
Positive	21	(30.4)
Negative	48	(69.6)
Preferred contraceptive; n (%)		
Intra-uterine device	4	(5.8)
Implanon	36	(52.2)
Injectable	24	(34.8)
Condom	5	(7.3)
Circumstances that led to sex work; n (%)		
Poverty and unemployment	64	(92.8)
Drug/alcohol abuse	5	(7.3)
First sexual encounter was voluntary; n (%)		
Yes	45	(65.2)
No	24	(34.8)

**Table 2 healthcare-11-01271-t002:** Obstetric history of the sex workers of North-West (NW) and Eastern Cape (EC) provinces from South Africa.

Variable	n (%)	
Ever Pregnant; n (%)		
Yes	62	(89.9)
No	7	(10.1)
Gravidity; n (%)		
0	7	(10.1)
1	24	(34.8)
2	23	(33.3)
3–5	15	(21.7)
Ever had an abortion; n (%)		
Yes	14	(20.3)
No	55	(79.7)
Have own children; n (%)		
Yes	60	(87.0)
No	9	(13.0)
Number of children; n (%)		
0	9	(13.0)
1	28	(40.6)
2	21	(30.4)
3–5	11	(15.9)

**Table 3 healthcare-11-01271-t003:** Practices and attitudes regarding condom use by South African sex workers.

Variable	n (%)	
Condom used in last sexual encounter; n (%)		
Yes	57	(82.6)
No	12	(17.4)
Frequency of condom use in last 3 encounters: n (%)		
Three times	43	(62.3)
Two or few times	26	(37.7)
Barriers to condom use; n (%)		
Yes	56	(81.2)
No	13	(18.8)
Advantages of using a condom; n (%)		
Prevents STIs	1	(1.5)
Prevents HIV infection	9	(13.0)
Prevents against pregnancy and diseases	59	(85.5)
Disadvantages of using a condom; n (%)		
Lessened pleasure	1	(1.5)
You won’t charge much	59	(85.5)
None	9	(13.0)
Motivation to use a condom; n (%)		
Protection	65	(94.2)
Because we are told	4	(5.8)
Discourages from using a condom; n (%)		
Clients refuse	25	(36.2)
Money paid by clients/acceptance	44	(63.8)
Factors influencing condom use; n (%)		
Prolonging life	8	(11.6)
Personal responsibilities	11	(15.9)
High rate of STIs and HIV	50	(72.5)
Risk of HIV infection if condom compliant; n (%)		
Unlikely	62	(82.9)
Less likely	7	(10.1)

**Table 4 healthcare-11-01271-t004:** Attitudes and practices regarding female condoms by South African sex workers.

Variable	n (%)	
Views on female condom; n (%)		
Attractive	1	(1.5)
Useful preventative method	21	(30.4)
Unattractive	13	(18.8)
Not comfortable	34	(49.3)
Female condom use in last 3 months; n (%)		
None	62	(89.9)
At least once	1	(1.5)
At least five times	6	(8.7)
Female condom use in last 12 months; n (%)		
None	53	(76.8)
At least once	12	(17.4)
At least five times	4	(5.8)

**Table 5 healthcare-11-01271-t005:** Perceptions towards female condoms by South African sex workers.

Variable	n (%)	
Female condom is better than male condom; n (%)		
Disagree	22	(31.9)
Neutral	41	(59.4)
Agree	6	(8.7)
Female condom is easy to insert; n (%)		
Disagree	26	(37.7)
Neutral	37	(53.6)
Agree	6	(8.7)
Female condoms are more expensive; n (%)		
Disagree	69	(100.0)
Neutral	0	(0)
Agree	0	(0)
Female condom use is against culture; n (%)		
Disagree	49	(71.0)
Neutral	6	(8.7)
Agree	14	(20.3)
Female condoms enhance pleasure; n (%)		
Disagree	0	(0)
Neutral	64	(92.8)
Agree	5	(7.3)
Female condoms are preferred by clients; n (%)		
Disagree	0	(0)
Neutral	58	(84.1)
Agree	11	(15.9)
Female condom shameful; n (%)		
Disagree	18	(26.1)
Neutral	48	(69.6)
Agree	3	(4.4)
Female condoms are available and accessible; n (%)		
Disagree	19	(27.5)
Neutral	3	(4.4)
Agree	47	(68.1)
Female condoms are well promoted; n (%)		
Disagree	7	(10.1)
Neutral	6	(8.7)
Agree	56	(81.2)
It is difficult to access female condoms; n (%)		
Very difficult	1	(1.5)
Somewhat difficult	7	(10.1)
Not difficult	61	(88.4)

**Table 6 healthcare-11-01271-t006:** Knowledge of selected categorical variables by South African sex workers.

Variable	Category	Prevalence		PR (95% CI)	*p*-Value
		n/N	(%)		
Barriers to condom use					
No		1/13	(7.7)	Reference	1
Yes		29/56	(51.8)	6.7 (1.01–45.0)	0.004
Province					
North-West		14/48	(29.2)	Reference	1
Eastern Cape		16/21	(76.2)	2.6 (1.6–4.3)	0.0003
HIV Status					
Positive		6/21	(28.6)	Reference	1
Negative		24/48	(50.0)	0.6 (0.3–1.2)	0.099

**Table 7 healthcare-11-01271-t007:** Bivariate Association of Knowledge Score and selected numerical variables (Linear Regression) for South African sex workers.

Variable	Co-Efficient	R^2^ (%)	*p*-Value
Age, years	−0.4	6.6	0.032
Average no. of daily clients	−1.9	13.3	0.002

## Data Availability

All data used in the study will be available from the corresponding.

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
