# Peer review of "Knowledge, Attitudes, and Practices of Sex Workers of Three South African Towns towards Female Condom Use and Contraceptives"

_healthcare, 2023, doi:10.3390/healthcare11091271_

Round 1

Reviewer 1 Report

Knowledge, Attitudes and Practices of sex workers of three South African towns towards female condom use and contraceptives

This study aimed to known about the acceptability and usage of female condoms and contraceptives among sex workers in South African. Many grammatical errors in the manuscript were observed. It needs reviewing carefully. I put other corrections in the manuscript.

Abstract section: a satisfactory conclusion and response to this manuscript's aim must be reached.

Introduction Section. This section is adequate.

Material and Methods Section. I do not know if the 69 people are representative of the total population of sex workers. You should calculate the number of people necessary for the data to be representative of the population. Put this information in the material and methods section.

Results Section. The results described are adequate.

Discussion section. The discussion is very long and is not punctual with the use of female and male condoms, so the conclusion is inadequate. You should comment more about acceptation or reprobation of condoms worldwide, as well as the advantages and disadvantages of the use.

Reference Section. Is need to review all references. Some of them need to be revised. Some of them require a web address and the date of the review.

Tables Section. You should modify all titles of tables because there is no information that the data obtained were of sex workers living in South Africa. The tables should give all information without having to read all manuscript.

Author Response

Conclusion The studies have confirmed low acceptability as manifested by low usage of female condoms. The study has reported an HIV prevalence of 30.4% among Brits and Grahamstown female sex workers. More than 60% of participants had completed their high school; a fifth had attained a tertiary qualification; the age of entry into sex work was 16 and the median sexual debut was 15 years. More than half of the participants were cohabit-ingcohabiting, and some were married. Most participants were not knowledgeable about the female condom and other prophylactic modalities used to prevent against STIs and unwanted pregnancies. However, the majority ofmost participants believed that even though the female con-doms were not expensive they were unattractive and were not easy to use/insert. As a resultresult, the use of female condoms was very low in this population. In general, condom use during transactional sex was not consistent as it tended to lower their in-come. Ironically, participants who reported barriers to condom use were 88% more likely to have adequate knowledge than those who did notnot, and this was statistically significant. Promotion of female condom use is needed, as well as education about the ad-vantages of using the female condom. Further research should be undertaken in this field. On a practical level, work has tomust be done to develop a female condom with features attractive to potential users.

please see attached file.

Reviewer 2 Report

The article by Sitonga el al reports the results of a survey assessing the knowledge, acceptability, usage, and attitudes toward female condoms and contraceptives among sex workers in three South African towns. 

In general, the study is well written and clear. However, there are some inconsistencies in some of the numbers reported throughout the text and omissions that could be addressed to improve the information related.

Specific comments:

Abstract: the abstract mainly reports results regarding condom use in general (mostly male condoms) but fails to describe specific results/insights from the study regarding use of female condoms. Please add this information and modify the conclusion to reflect what it was learned from the study.

There are several instances where the numbers stated in the text do not match the numbers on the tables. Please correct the following:

·         Abstract, line 29 states entry into sex work was 16 years but table 1 states 22.8 years

·         Results, line 176: Text says range of age is 18-35 but table 1 says 18-46

·         Line 193 mentions Table 3 but should be Table 2

·         Discussion, Line 242 states 69.6% participants live with HIV but should say 30.4%

·         Line 265 states 30.5% of participants had adequate knowledge but above and in Results says 43.5%.

·         Line 380 states age of entry into sex work was 16 years but table 1 says 22.8 years.

·         In addition, lines 191-192 seem to be truncated and missing information

·         Line 304: it is not clear what percentages are being reported.

Other comments:

It would be informative if the authors described in the introduction what type of female condoms are available in the locations surveyed and what are the accessibility options to sex workers or women in general (price range compared to male condoms, availability in pharmacies, health facilities or other, etc.) so the reader can understand how these barriers may influence use of female condoms in addition to perceptions women have?  

In general, the discussion repeats many of the results reported in the results section without discussing possible reasons for the observations or potential opportunities for intervention. Examples below:

Lines 265-268 in discussion repeat results. Can the authors comment instead on possible reasons why younger participants had more knowledge than older participants and what suggestions do the authors have to better inform older women?

Line 284: The authors state the women had a higher level of education compared to other groups in South Africa and this contradicts the association of sex work and lack of education previously described by others. How do the authors explain these observations? The women interviewed reported beginning sex work due to poverty or unemployment – does this reflect fewer opportunities for employment in these rural areas despite their higher level of education?  Also, please modify sentence in lines 288-289 as is not clear as written.

Why do the authors think the HIV prevalence was lower in this group compared to other groups of sex workers in South Africa or reported by the United Nations?

The authors describe use of female condoms as empowering by removing some of the barriers presented by regular (male) condoms, but they do not elaborate on the specific objections the women had to the use of female condoms, besides mentioning they are perceived as uncomfortable, unattractive, and not easy to use.  Can the authors comment on whether promotion (or lack of) is failing at providing accurate information and what strategies can be used to change these perceptions. Also, what is being done in the field to gather input on desirable features attractive to potential users of female condoms and the development of new, more acceptable options.

Author Response

Abstract: Background: Female sex workers are a marginalised and highly vulnerable population who are at risk of HIV and other sexually transmitted diseases, harassment, and unplanned pregnancies. However, little is known about the acceptability and usage of female condoms and contraceptives among sex workers in small South African towns like Brits, Rustenburg, and Grahamstown. We further explore factors associated with knowledge of female condoms and other HIV prophylactic modalities. There 2 types of female condoms Cupid and FC2 that are available and accessible in these two provinces, they are distributed at brothels, sheebens, night clubs, and in all health facilities. Methods: This study is a quantitative study design with both descriptive and analytical components, undertaken in September 2018. A validated, researcher-administered questionnaire adapted from the South African Youth Risk Behaviour Survey was used. Data were analysed using STATA 14.1. Normally distributed numerical data are summarised using the mean, standard deviation and range. Numerical data that were not normally distributed are summarised using the median and interquartile range. Simple linear regression is used to compare two numerical variables. The prevalence ratio (PR) is used to determine the association of two categorical variables. The level of significance is p-value<0.05. Results: Sixty-nine (69) participants are reported on in this study which recorded an HIV prevalence of 30.4% (n = 21/69). A high number of participants had completed high school (60.9%); the median age of entry into sex work was 16 and the median sexual debut was 15 years. More than half of the participants (37/69 or 53.6%) were cohabiting. In general, condom use during transactional sex was not consistent as it affected their income. Participants who reported barriers to condom use were 90% more likely to have adequate knowledge than those who did not, and this was statistically significant (PR = 1.9; p-value<0.0001). Conclusion: Sex workers, like all members of society have a right to access to promotive, preventive and curative healthcare and should thus have their health needs known and addressed. This study is one of few such efforts for small rural towns in South Africa.

Reviewer 3 Report

- Knowledge and attitudes towards the use of contraception in areas with a high HIV prevalence. If carried out properly, can prevent the spread of HIV?

- Controls for sex workers are ineffective because it is clear that few women in general use them.

- Along with assessment of KAP, health education regarding contraception in particular female condom usage may have added value to the study, but, the objectives would be different and it would be an interventional study.

-Only 17.4% practiced female condom though more than 88% answered that acecess to it is easy and more than 50% do not know whether the usage is difficult or easy. Their preferred contraceptive is Implanon and injectables which are easy to practice but do not prevent transmission of HIV which has long term effects on their health and reduces their survival if they contract the disease. Even if they prefer injectable contraceptives the concept of double contraception should be promoted to reduce HIV transmission.

Reviewer 4 Report

The paper entitled: “Knowledge, Attitudes and Practices of sex workers of three South African towns towards female condom use and contraceptives” aims to determine the knowledge, attitudes, and practices of female sex workers in North West and Eastern Cape provinces of South Africa on the use of female condoms and contraceptives.

At the methodological level, the authors surveyed 70 participants over 10 days between the 20th and 29th of September 2018. The study was conducted in two brothels of Brits and Rustenburg (North West province); and in two brothels of Joza township in Grahamstown (Eastern Cape province), two mining towns with high proportions of migrant male workers.

The sample researched is very small. Therefore, the findings of this study should be considered with caution. However, the population under study is difficult to access. Most studies involving interviews with sex workers are carried out ex-situ (id est, in NGO’s shelters, jails, etc.). However, this study was carried out in situ. In my opinion, studies carried out in situ are more reliable, because participants’ discourse interviewed ex-situ is more contaminated. On the other hand, there are only a few studies reporting female condom use by sex workers and the acceptability and usage of the female condom among sex workers in South Africa. This study identifies the health needs of sex workers in rural mining towns with high proportions of truck drivers and migrant workers who work far from their wives and demand the services of sex workers.

According to the authors, the high HIV prevalence among South African sex workers results from the illegality of sex work in South Africa. Therefore, advocacy for the sexual and reproductive health of sex workers is limited. Neo-abolitionist feminist discourse has argued that sex work is essentially violence against women. Therefore, sex work should be wiped out. Under the neo-abolitionist paradigm, there is no room for the advocacy for the sexual and reproductive health of sex workers; because sex workers should be persuaded to exit sex work. Neo-abolitionism is the predominant paradigm in relation to sex work. Most legislation and policies on sex work around the world, and South Africa is not an exception, follow the neo-abolitionist paradigm.

I would like the authors to discuss in one or two pages the implications of neo-abolitionist South African policies in the difficulties to attend sexual and reproductive health of sex workers in South Africa.

In this study, the average age of participants was 32 years. According to the authors “the average age of sex workers was between 25 and 35 years in the continent” (p. 10). This information is misleading.  The average age of sex workers who participate in studies is significantly higher than the average age of the total population of sex workers because younger sex workers are less willing to participate in studies.

I want the authors to discuss more in-depth about the interviewees' age at entry into sex work. The authors point out in the abstract that “the median age of entry into sex work was 16”. How many interviewees began commercial sex work before 18 years of age? for how long interviews have been involved in sex work?

Finally, the references should be updated. There are no references to papers published in 2021 & 2023, and there is only a reference to an article published in 2022.  
